Low-cost, low-input RNA-seq protocols perform nearly as well as high-input protocols

Combs Peter A. 1 peter.combs@berkeley.edu
Eisen Michael B. 2 3
1 Graduate Program in Biophysics, University of California , Berkeley, CA , USA
2 Department of Molecular and Cell Biology, University of California , Berkeley, CA , USA
3 Howard Hughes Medical Institute, University of California , Berkeley, CA , USA
Wilke Claus
Electronic publication date: 2015 Mar 26
Publication date: 2015
Volume: 3
Electronic Location ID: e869
Received 2015 Jan 6; Accepted 2015 Mar 11
Copyright: © 2015 Combs and Eisen
Copyright year: 2015
Copyright holder: Combs and Eisen
License: This is an open access article distributed under the terms of the Creative Commons Attribution License, which permits unrestricted use, distribution, reproduction and adaptation in any medium and for any purpose provided that it is properly attributed. For attribution, the original author(s), title, publication source (PeerJ) and either DOI or URL of the article must be cited.
License URL: https://creativecommons.org/licenses/by/4.0/

Keywords: RNAseq, Single cell, Protocols, Linear output

Funding: Howard Hughes Medical Institute investigator award National Institutes of Health training #T32 HG 00047 NIH S10 Instrumentation S10RR029668 S10RR027303 This work was funded by a Howard Hughes Medical Institute investigator award to Michael B. Eisen, and National Institutes of Health training grant #T32 HG 00047 to PAC. This work used the Vincent J. Coates Genomics Sequencing Laboratory at UC Berkeley, supported by NIH S10 Instrumentation Grants S10RR029668 and S10RR027303. The funders had no role in study design, data collection and analysis, decision to publish, or preparation of the manuscript.

==============================
Recently, a number of protocols extending RNA-sequencing to the single-cell regime have been published. However, we were concerned that the additional steps to deal with such minute quantities of input sample would introduce serious biases that would make analysis of the data using existing approaches invalid. In this study, we performed a critical evaluation of several of these low-volume RNA-seq protocols, and found that they performed slightly less well in per-gene linearity of response, but with at least two orders of magnitude less sample required. We also explored a simple modification to one of these protocols that, for many samples, reduced the cost of library preparation to approximately $20/sample.

Introduction

Second-generation sequencing of RNA (RNA-seq) has proven to be a sensitive and increasingly inexpensive approach for a number of different experiments, including annotating genes in genomes, quantifying gene expression levels in a broad range of sample types, and determining differential expression between samples. As technology improves, transcriptome profiling has been able to be applied to smaller and smaller samples, allowing for more powerful assays to determine transcriptional output. For instance, our lab has used RNA-seq on single Drosophila embryos to measure zygotic gene activation (Lott et al., 2011) and medium-resolution spatial patterning (Combs & Eisen, 2013). Further improvements will allow an even broader array of potential experiments on samples that were previously too small.

For instance, over the past few years, a number of groups have published descriptions of protocols to perform RNA-seq on single cells (typically mammalian cells) (Tang et al., 2009; Ramsköld et al., 2012; Sasagawa et al., 2013; Hashimshony et al., 2012; Islam et al., 2011). A number of studies, both from the original authors of the single-cell RNA-seq protocols and from others, have assessed various aspects of these protocols (such as the lower limit of detection, strand specificity, and uniformity of coverage), both individually and competitively (Levin et al., 2010; Bhargava et al., 2014; Wu et al., 2014; Marinov et al., 2013). One particularly powerful use of these approaches is to sequence individual cells in bulk tissues, revealing different states and cellular identities (Buganim et al., 2012; Treutlein et al., 2014).

However, we felt that published descriptions of single-cell and other low-volume protocols did not adequately address whether a change in concentration of a given transcript between two samples would result in a proportional change in the FPKM (or any other measure of transcriptional activity) between those samples. While there are biases inherent to any protocol, we were concerned that direct amplification of the mRNA would select for PCR compatible genes in difficult to predict, and potentially non-linear ways. For many of the published applications of single cell RNA-seq, this is not likely a critical flaw, since the clustering approaches used are moderately robust to quantitative changes. However, to measure spatial and temporal activation of genes across an embryo, it is important that the output is monotonic with respect to concentration, and ideally linear. A linear response allows for more easily interpretable experimental results, without necessarily relying on complicated transformations of the data.

While it is possible to estimate absolute numbers of cellular RNAs from an RNA-seq experiment, doing so requires spike-ins of known concentration and estimates of total cellular RNA content (Mortazavi et al., 2008; Lin et al., 2012). However, many RNA-seq experiments do not do these controls, nor are such controls strictly necessary under reasonable, though often untested, assumptions of approximately constant RNA content. While ultimately absolute concentrations will be necessary to fully predict properties such as noise tolerance of the regulatory circuits (Gregor et al., 2007; Gregor et al., 2005), many current modeling efforts rely only on scaled concentration measurements, often derived from in situ-hybridization experiments (Garcia et al., 2013; Ilsley et al., 2013; He et al., 2010). Given that, we felt it was not important that different protocols should necessarily agree on any particular expression value for a given gene, nor are we fully convinced that absolute expression of any particular gene can truly reliably be predicted in a particular experiment.

In order to convince ourselves that data generated from limiting samples would be suitable for evaluating the spatial distribution of gene expression or other experiments where a linear response is necessary for proper interpretation of the data, we evaluated several protocols for performing RNA-seq on extremely small samples. We also investigated a simple modification to one of the protocols that reduced sample preparation cost per library by more than 2-fold. This study provides a single, consistent comparison of these diverse approaches, and shows that in fact all data from the low-volume protocols we examined are usable in similar contexts to the earlier bulk approach.

Methods

RNA extraction, library preparation, and sequencing

We performed RNA extraction in TRIzol (Life Technologies, Grand Island, New York, USA) according to manufacturer instructions, except with a higher concentration of glycogen as carrier (20 ng) and a higher relative volume of TRIzol to the expected material (1 mL, as in Lott et al. (2011) and Combs & Eisen (2013)). We quantified RNA concentrations using a fluorometric Qubit RNA HS assay (Life Technologies, Grand Island, New York, USA).

TruSeq libraries were prepared with the “TruSeq RNA Sample Preparation Kit v2” (Illumina Cat.#RS-122-2001) according to manufacturer instructions, except for the following modifications. All reactions were performed in half the volume of reagents. We find that this increases the effective concentration of RNA and cDNA. We performed all reactions and cleanups in 8-tube PCR strip tubes, which allowed us to reduce the volume of Resuspension Buffer to minimize volume left behind after each cleanup.

Clontech libraries were prepared with the “Low Input Library Prep Kit” (Clontech Cat.#634947). We generated cDNA by using TruSeq reagents until the cDNA synthesis step. Then, we used the Low Input Library Prep Kit to modify the cDNA into sequencing-competent libraries. We assume that a similar cDNA synthesis could be performed using oligo dT Dynabeads, RNA fragmentation reagents, and Superscript II (Life Technologies, Grand Island, New York, USA), for an approximate cost per sample of $15, but have not directly tested this with the Clontech reagents.

TotalScript libraries were prepared with the “TotalScript RNA-Seq Kit” and “TotalScript Index Kit” (Epicentre Cat.#TSRNA1296 and TSIDX12910). We followed the manufacturer’s instructions, and used the oligo dT priming option. We performed the mixed priming option in parallel, which yielded approximately 4-fold more library, but did not sequence them due to concerns of ribosomal contamination.

SMARTseq2 libraries were prepared according to the protocol in Picelli et al. (2014). Because we had already extracted and mixed the RNA, we began at step 5 with 3.7 µL of dNTPs and 1 µL of 37 µM oligo dT primer, yielding the same concentration of primer and oligo as originally reported. We used 18 cycles for the preamplification PCR in step 14, added 1 ng of cDNA to the Nextera XT reactions in step 28, and used 6 and 8 cycles for the final enrichment in step 33 (experiments 2 and 3, respectively).

Libraries were quantified using a combination of Qubit High Sensitivity DNA (Life Technologies, Grand Island, New York, USA) and Bioanalyzer (Agilent Technologies, Sunnyvale, California, USA) readings. Total yield Y in femtomoles was estimated using Qubit concentration C measured in (ng/µL), total volume V in µL, average size S in bp, (1) Yfmole=Cng/µL⋅10−9[g/ng]⋅VµL⋅1015fmole/mole÷608.9g/mole÷S[bp].

We then pooled libraries to equalize index concentration before sequencing.

Due to a pooling error in experiment 2 where non-concentration normalized tubes were mistakenly used instead of the normalized samples, the TruSeq libraries were included at much higher abundance. Pooled libraries were then submitted to the Vincent Coates Genome Sequencing Laboratory for 50bp single-end sequencing according to standard protocols for the Illumina HiSeq 2500. Bases were called using HiSeq Control Software v1.8 and Real Time Analysis v2.8.

Mapping and quantification

Reads were mapped using STAR (Dobin et al., 2012) to a combination of the FlyBase reference genome version 5.54 for D. melanogaster and D. virilis (McQuilton, St. Pierre & Thurmond, 2012). We randomly sampled the mapped reads to use an equal number in each sample compared. We used HTSeq (command line options htseq-count --idattr='gene_name' --stranded=no --sorted=pos) to count absolute read abundance per gene (Anders, Pyl & Huber, 2014). All custom analysis software is available at https://github.com/eisenlab/SliceSeq, and is primarily written in Python (Cock et al., 2009; Hunter, 2007; Jones et al., 2001; Perez & Granger, 2007). Commit c6b3d3e was used to perform all analyses in this paper.

Simulation of Experiment 2

We wrote a Python script that simulated Experiment 2 assuming only uncorrelated counting noise in the number of reads per gene. The read counts from the sample with 20% D. virilis and the TruSeq protocol was used to generate the base probabilities. D. virilis gene probabilities were adjusted downwards, and the remaining probability was assigned evenly to the D. melanogaster genes. The SciPy function stats.multinomial was used to simulate read counts, assuming an equal number of reads as in the original experiment. Gene expression levels were normalized using Eq. (2), as in the actual experiment.

Results

Experiment 1: evaluation of Illumina TruSeq

In our hands, the Illumina TruSeq protocol has performed extremely reliably with samples on the scale of 100 ng of total RNA, the manufacturer recommended lower limit of the protocol. However, attempts to create libraries from much smaller samples yielded low complexity libraries, corresponding to as much as 30-fold PCR duplication of fragments (Text S1). Anecdotally, less than 5% of libraries made with at least 90 ng of total RNA yielded abnormally low concentrations, which we observed correlated with low complexity (Data not shown). To determine the lower limit of input needed to reliably produce libraries, we attempted to make libraries from 40, 50, 60, 70, and 80 ng of Drosophila total RNA, each in triplicate. Yields are shown in Table 1.

Table 1 Total TruSeq cDNA library yields made with a given amount of input total RNA.

Yields measured by Nanodrop of cDNA libraries resuspended in 25 µL of EB. The italicized samples were near the lower limit of detection, and when analyzed with a Bioanalyzer, showed abnormal size distribution of cDNA fragments.

Amount input RNA	Replicate A	Replicate B	Replicate C	
40 ng	57 ng	425 ng	672 ng	
50 ng	435 ng	768 ng	755 ng	
60 ng	115 ng	663 ng	668 ng	
70 ng	300 ng	593 ng	653 ng	
80 ng	468 ng	550 ng	840 ng	

We considered the two libraries with lower than usual concentration to be failures. Although there is detectable material post-amplification, the size distributions as measured by Bioanalyzer of these libraries is significantly different than known good libraries and manufacturer provided documentation (Fig. S1). In our experience, sequencing libraries with much lower than usual yield and abnormal size distributions has yielded libraries with low complexity and poor correlation to replicates.

While a failure rate of approximately 1 in 3 might be acceptable for some purposes, we ultimately wanted to perform RNA sequencing on precious samples, where a failure in any one of a dozen or more libraries would necessitate regenerating all of the libraries. Furthermore, due to the low sample volumes involved (less than approximately 500 pg of poly-adenylated mRNA), common laboratory equipment is not able to determine the particular point in the protocol where the failures occurred.

It is clearly possible to use less than the manufacturer suggested amount of input RNA. Thus, we consider 70 ng of total RNA to be the conservative lower limit to the protocol. While this is about 30% smaller than the manufacturer suggests, it is still several orders of magnitude larger than we needed it to be. We therefore considered using other small-volume and “single-cell” RNA-seq kits, which often use a pre-amplification step that is known to influence estimation of absolute levels (Picelli et al., 2013).

Experiment 2: competitive comparison of low-volume RNAseq protocols

We first sought to determine whether the low-volume RNAseq protocols available faithfully recapitulate linear changes in abundance of known inputs, even if absolute levels are not directly comparable to other protocols. We generated synthetic spike-ins by combining D. melanogaster and D. virilis total RNA in known, predefined proportions of 0, 5, 10, and 20% D. virilis RNA. For each of the low-volume protocols, we used 1 ng of total RNA as input, whereas for the TruSeq protocol we used 100 ng.

Although pre-defined mixes of spike-in controls have been developed and are commercially available (Jiang et al., 2011), we felt it was important to ensure that a given protocol would function reproducibly with natural RNA, which almost certainly has a different distribution of 6-mers, which could conceivably affect random cDNA priming and other amplification effects. Furthermore, our spike-in sample more densely covers the approximately 105 fold coverage typical of RNA abundances. It should be noted, however, that our sample is not directly comparable to any other standards, nor is the material of known strandedness. We assumed that the majority of each sample is from the standard annotated transcripts, but did not verify this prior to library construction and sequencing.

We then estimated yield by measuring concentration in ng/µL with a Qubit High Sensitivity DNA assay and average fragment size with a Bioanalyzer High Sensitivity chip. The different protocols had a variation in yield of libraries from between 6 fmole (approximately 3.6 trillion molecules) and 2,400 femtomoles, with the TruSeq a clear outlier at the high end of the range, and the other protocols all below 200 fmole (Table 3). While the number of PCR cycles in the final enrichment steps can be adjusted, all of these quantities are sufficient to generate hundreds of millions of reads—far more than is typically required for an RNA-seq experiment. We pooled the samples, attempting equimolar fractions in the final pool; however, due to a pooling error, we generated significantly more reads than intended for the TruSeq protocol, and correspondingly fewer in the other protocols. Unless otherwise noted, we therefore sub-sampled the mapped reads to the lowest number of mapped reads in any library in order to provide a fair comparison between protocols.

We were interested in the fold-change of each D. virilis gene across the four libraries, rather than the absolute abundance of any particular gene. Therefore, after mapping and gene quantification, we normalized the abundance Aij of every gene i across the j = 4 libraries by a weighted average of the quantity Qj of D. virilis in library j, as show in Eq. (2). Thus, within a given gene, a linear fit of Aˆij vs. Qj should have a slope of one and an intercept of zero. As expected, this normalized abundance increased with increasing D. virilis concentration (Fig. S2). (2) Aˆij=Aij÷∑jQjAij∑jQj2.

We then filtered the D. virilis genes for those with at least 20 mapped fragments in the library with 20% D. virilis, then calculated an independent linear regression and the Pearson correlation coefficient between the expected and measured concentration of D. virilis for each of those genes. As shown in Fig. 1A and Fig. S3, this can be thought of as plotting the measured and known values, then fitting a line for each gene. As expected, for every protocol, the mean slope was 1 (t-test, p < 5 × 10−7 for all protocols). Similarly, the average intercepts for all protocols was 0 (t-test, p < 5 × 10−7 for all protocols). Also unsurprisingly, the TruSeq protocol had a noticeably higher mean correlation coefficient (0.98 ± 0.02) than any of the other protocols (0.95 ± 0.06, 0.92 ± 0.09, and 0.95 ± 0.06 for Clontech, TotalScript, and SMART-seq2, respectively). The mean correlation coefficient was statistically and practically indistinguishable between the Clontech libraries and the SMART-seq2 libraries (t-test p = .11, Fig. 1). Taken together, all of these measures indicate that the TruSeq protocol is better able to capture the linear trend in increasing transcript number.

Figure 1 Comparison of linearity between different RNA-seq protocols.

(A) Normalized levels of gene expression Â across libraries using the TruSeq protocol, where each line is for a different gene. (B–E) Distributions of slopes, intercepts, and correlation coefficient for linear regressions of the abundance of each gene, as in (A).

While the TruSeq protocol clearly performed better than the low-volume kits (Fig. 1), we wondered how well an ideal RNA-seq protocol could perform. We simulated an experiment with known levels of D. virilis spike in and assuming a multinomial distribution of read counts, and repeated the simulation 1,000 times to estimate the distribution of relevant quality metrics (Fig. S4). Surprisingly, the mean correlation coefficient for the TruSeq protocol was higher than the mean correlation coefficient of every repetition of the simulation, though indistinguishable for practical purposes (0.984 vs. 0.982). The slopes were equally well clustered around 1, with an interquartile range of 0.0864 for the TruSeq protocol compared to 0.0843, the mean of all simulations; 13% of simulations had a higher IQR. We thus conclude that the major limiting factor for the TruSeq protocol to generate a linear response in the data is likely the sequencing depth, whereas the other protocols all contain additional biases.

Although there is some variation in the precise shape of the distributions of fit parameters, these were relatively small compared to the difference between any of these and the conventional TruSeq protocol. Indeed, the major differentiator we found among the low-volume protocols we compared was cost. For only a handful of libraries, the kit-based all inclusive model of the Clontech and TotalScript kits could be a significant benefit, allowing the purchase of only as much of the reagents as required. By contrast, the Smart-seq2 protocol requires the a la carte purchase of a number of reagents, some of which are not available or more expensive per unit for smaller quantities. Furthermore, there could potentially be a “hot dogs and buns” problem, where reagents are sold in non-integer multiples of each other, leading to leftovers. Many of these reagents are not single-purpose, however, so leftovers could in principle be repurposed in other experiments.

Table 2 Summary of protocols used in experiments 2 and 3.

Cost is estimated per library assuming a enough libraries to consume all reagents at US catalog prices as of May 2014, and includes $2 for TRIzol RNA extraction, but not experimenter labor, sample QC, labware, or sequencing. Difference in prices in the Smart-seq2 protocols entirely due to scaling in cost of Nextera reagents.

Protocol	Shorthand	Cost/library	
TruSeq	TruS	$45	
Clontech	CT	$105	
Epicentre TotalScript	TotS	$115	
Smart-seq2, standard protocol	SS	$55	
Smart-seq2, 2.5 fold dilution	SS—2.5×	$28	
Smart-seq2, 5 fold dilution	SS—5×	$20	

Table 3 Sequencing summary statistics for libraries.

Protocols are the shorthands used in Table 2. Reads indicates the total number of reads, and Mapped the total number of reads that mapped at least once to either genome. Experiments 2 and 3 were run in a single HiSeq lane each. Yield estimates were generated by adjusting Qubit High Sensitivity DNA readings by the average fragment size as measured by Bioanalyzer.

Expt	Protocol	% D. virilis	Yield	Total reads	Mapped reads	
2	CT	0%	6.5 fmole	3,803,843	3,374,520 (89%)	
2	”	5%	15.7 fmole	4,372,738	4,164,781 (95%)	
2	”	10%	47.4 fmole	10,013,087	9,527,023 (95%)	
2	”	20%	17.8 fmole	4,781,463	4,317,101 (90%)	
2	TotS	0%	176.8 fmole	3,281,134	2,930,058 (89%)	
2	”	5%	170.2 fmole	2,498,134	2,237,330 (90%)	
2	”	10%	102.5 fmole	5,777,523	5,424,366 (94%)	
2	”	20%	119.9 fmole	6,068,996	5,740,496 (95%)	
2	TruS	0%	2,401.0 fmole	67,560,511	64,024,881 (95%)	
2	”	5%	2,001.1 fmole	23,370,854	22,589,083 (97%)	
2	”	10%	2,174.2 fmole	39,454,390	38,093,763 (97%)	
2	”	20%	2,379.2 fmole	35,265,536	34,304,792 (97%)	
2	SS	0%	34.3 fmole	2,439,518	2,297,087 (94%)	
2	”	5%	59.6 fmole	2,550,023	2,419,889 (95%)	
2	”	10%	67.9 fmole	2,534,628	2,444,568 (96%)	
2	”	20%	39.8 fmole	2,504,340	2,389,850 (95%)	
3	SS—2.5×	0%	104.4 fmole	15,769,915	14,393,959 (91%)	
3	”	1%	124.7 fmole	21,349,748	20,084,131 (94%)	
3	”	5%	113.0 fmole	17,047,120	16,329,641 (96%)	
3	”	10%	103.5 fmole	23,762,232	22,372,562 (94%)	
3	”	20%	123.8 fmole	20,809,781	20,041,548 (96%)	
3	SS—5×	0%	59.4 fmole	19,214,155	17,324,598 (90%)	
3	”	1%	58.6 fmole	23,832,274	22,364,220 (94%)	
3	”	5%	65.4 fmole	18,149,452	17,157,450 (95%)	
3	”	10%	28.8 fmole	15,821,419	14,869,864 (94%)	
3	”	20%	57.2 fmole	22,466,345	21,620,603 (96%)	

Table 4 Distribution of fit parameters.

A simple linear fit, Aˆij=mi⋅Qj+bi, was computed for each gene i and a correlation coefficient r was calculated. For brevity, x¯ is the mean of some variable x, and σx is its standard deviation.

Protocol	m¯±σm	b¯±σb	r¯±σr	
TruSeq	1.01 ± 0.0698	−0.108 ± 1.05	0.98 ± 0.019	
Clontech	1.01 ± 0.12	−0.217 ± 1.79	0.95 ± 0.061	
Epicentre TotalScript	0.952 ± 0.129	0.715 ± 1.93	0.93 ± 0.094	
Smart-seq2	1.03 ± 0.121	−0.506 ± 1.82	0.95 ± 0.057	
Smart-seq2, 2.5 fold dilution	0.996 ± 0.111	0.0623 ± 1.67	0.96 ± 0.053	
Smart-seq2, 5 fold dilution	1.01 ± 0.111	−0.173 ± 1.66	0.96 ± 0.049	

Experiment 3: further modifications to the SMART-seq2 protocol

Although the SMART-seq2 was the cheapest of the protocols when amortized over a large number of libraries, we wondered whether it could be performed even more cheaply without compromising data quality. This would enable us to include more biological replicates in the future experiments for which we are evaluating these protocols. In the original protocol, we noticed that roughly 60% of the cost came from the Nextera XT reagents. Thus, reducing the cost of tagmentation was the obvious goal to target.

We made additional libraries, again starting with 1ng of total RNA. We amplified a single set of spike-in libraries with 0, 5, 10, and 20% D. virilis total RNA as in experiment 2, and made a single an additional sample with 1% D. virilis RNA. Starting at the point in the SMART-seq2 protocol where tagmentation was started, we performed reactions in volumes 2.5 × and 5 × smaller, using proportionally less cDNA as well. Due to the low total yield, we increased the number of enrichment cycles from 6 to 8 (see ‘Methods’).

When normalized to the same number of reads as in experiment 2, the protocols with diluted Nextera reagents performed effectively identically: for instance, the mean correlation coefficients were in both cases 0.96 ± 0.05 (Fig. 2 and Table 4). This is despite the additional cycles of enrichment, which improved yield.

Figure 2 Distributions of slopes, intercepts, and correlation coefficients for experiment 3.

Nextera XT reactions were reduced in volume by the indicated amount.

Because we used a common set of pre-amplified cDNA samples that was performed in a distinct pre-amplification from experiment 2, we can estimate the contribution of that pre-amplification to the overall variation. If, in fact, the pre-amplification is a major contributor to the variation, then we would expect to find that the correlation between, for instance, the slopes of two runs of the same experiment with different pre-amplifications would be significantly lower than the correlation between the slopes of two runs using the same pre-amplified cDNA pools.

Unsurprisingly, the sets of samples that used the same preamplification were more correlated with each other than with the set of samples that used a separate pre-amplification (Fig. 3). By analogy to dual-reporter expression studies such as Elowitz et al. (2002), we term variation along the diagonal “extrinsic noise” (ηext = std(m1 + m2)), and variation perpendicular to the diagonal “intrinsic noise” (ηint = std(m1 − m2)), being intrinsic to the pre-amplification step. Using that metric, the intrinsic noise is lower for the samples with the same pre-amplification (ηint = 0.09) than for the samples with different pre-amplifications (ηint = 0.16). Somewhat surprisingly, the extrinsic noise is higher for the samples with the same pre-amplification (ηext = 0.20 vs. ηext = 0.16), perhaps due to the 2 additional cycles of PCR enrichment.

Figure 3 Estimating the source of preamplification noise.

Plotted are the estimated slopes for each gene between experiments. The blue, “Different pre-amplification” compares the 2.5× diluted and full sized reactions, whereas the green “same pre-amplification” points compare the 2.5× and 5× dilution samples, which used the same pre-amplified cDNA but different tagmentation reactions.

Discussion

When sample size is not the limiting factor, it is clear that using well-established protocols that involve minimal sequence-specific manipulation of the sample yields the best results, both in terms of reproducibility and linearity of response. However, if it is not practical to collect such relatively large samples, experiment 2 shows that any of the “single-cell” protocols we have tested should perform similarly to each other, and can be used as a drop-in replacement. While preamplification steps do introduce some detectable variance, it is not vastly detrimental to the data quality, and does not introduce obvious sequence-specific biases.

Such methods should be strongly preferred if it is feasible to collect a suitably homogenous sample. While bulk tissues may be a mixture of multiple distinct cell types, this may or may not affect the particular research question an RNAseq experiment is designed to answer. In our hands, the lower limit of reliable library construction using the Illumina TruSeq kit is approximately 70 ng of total RNA and we have used this amount of RNA in as-yet unpublished experiments on dissected slices of embryos. With non precious samples, the practical limit is likely to be even lower. The manufacturer suggested 100 ng is almost certainly safe, and we can think of relatively few experiments where it is not practical to collect more RNA than this. Although we have anecdotally observed significant user-to-user variation within our lab, it seems unreasonable to expect order-of-magnitude improvements are possible in techniques for precious samples. We suggest that this limit may be related to cDNA binding to tubes or purification beads, but since the quantities are lower than the detection threshold of many standard quality control approaches, we cannot directly verify this.

Compared to the regimes these protocols were designed for, we used a relatively large amount of input RNA—1 ng of total RNA—corresponding to approximately 50 nuclei of a mid-blastula transition Drosophila embryo. Previous studies have shown that this amount of RNA is well above the level where stochastic variation in the number of mRNAs per cell will strongly affect the measured expression of a vast majority of genes (Marinov et al., 2013). It is nevertheless a small enough quantity to be experimentally relevant. For instance, we have previously dissected single embryos into approximately 12 sections, yielding approximately 10 ng per section (Combs & Eisen, 2013), and one could conceivably perform similar experiments on imaginal discs or antennal structures, which contain a similar amount of cells (Klebes et al., 2002; Hansson & Anton, 2000).

One of the more striking results is that costs can be significantly reduced by simply performing smaller reactions, without noticeably degrading data quality. We do not suspect this will be true for arbitrarily small samples, such as from single cells. Instead, it is likely only true for samples near the high end of the effective range of the protocol. We have not explored where this result breaks down, and strongly caution others to verify this independently using small pilot experiments before scaling up.

Conclusions

The selection of protocols for performing RNAseq depends on the amount of material available to be profiled. We found that high quality libraries can be generated with slightly less than the manufacturer’s recommended minimum using the standard Illumina TruSeq protocol. When sufficient material is available to use the TruSeq protocol, we find that this produces data with a better linear response to the increasing concentration of any given gene than a number of different “single cell” protocols, which have roughly comparable performance in this metric. Finally, we found that at least one of these protocols, SMARTseq2 could be easily modified to significantly reduce the cost of library preparation, without compromising data quality.

Supplemental Information

Text S1 Estimating complexity of RNAseq libraries

Click here for additional data file.

Supplemental Information 1 Representative Bioanalyzer traces of libraries with both good and abnormal size distributions

Click here for additional data file.

Supplemental Information 2 Histograms of distributions of expression levels for all the protocols we used

Increasing the amount of D. virilis in each library increases the average abundance of D. virilis genes.

Click here for additional data file.

Supplemental Information 3 Similar to Fig. 1A, we have plotted the abundance of each gene individually for all of the protocols we used

Click here for additional data file.

Supplemental Information 4 Fit parameters for one iteration of a multinomial simulation of Experiment 2

The distributions are practically indistinguishable from the corresponding distribution for the TruSeq data, in 1B.

Click here for additional data file.

We would like to thank Lior Pachter for suggesting the simulation experiment, and reviewers Angela Wu and Scott Hunicke-Smith for their detailed comments and criticisms.

Additional Information and Declarations

Competing Interests

Author Contributions

Data Deposition

The authors declare there are no competing interests.

Peter A. Combs conceived and designed the experiments, performed the experiments, analyzed the data, contributed reagents/materials/analysis tools, wrote the paper, prepared figures and/or tables, reviewed drafts of the paper.

Michael B. Eisen conceived and designed the experiments, contributed reagents/materials/analysis tools, reviewed drafts of the paper.

The following information was supplied regarding the deposition of related data:

Gene Expression Omnibus GSE64673

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
