# Peer review of "Low-cost, low-input RNA-seq protocols perform nearly as well as high-input protocols"

_PeerJ, doi:10.7717/peerj.869_

## Round 0.1 · original submission · Minor Revisions

· Academic Editor

Minor Revisions

Both reviewers liked the work overall but had some comments, in particular regarding the presentation of the work. Reviewer 1 also had extensive comments regarding figures.

·

Basic reporting

In general the article is well-written, clear and to the point. But this part could use more clarification: “The TruSeq protocol had a noticeably higher mean correlation coefficient than any of the other protocols” Some additional description of how the correlation coefficient was computed would be helpful here. It was not immediately clear to me whether this correlation coefficient was computed between technical replicates of the same experiment (i.e. correlation of the linear regression over all spike in concentrations) or between different experiments (i.e. correlation of the gene expressions over all genes between experiments using different spike-in concentrations). In addition, how was the mean correlation coefficient obtained? Was it by computing the coefficient over all pair-wise combinations, then taking the average? Or was it the mean over some randomly chose pairs?

The graphics and figures, however, need substantial work. First, the labels on the graphs in Figure 1 are not very descriptive or clear. Figure 1A axis label “normalized expression” needs to be more specific, either giving units, or have a description in the figure caption describing what this is, even though it is in the text it would make it easier for readers taking a quick look at the article to understand what is being plotted. Second, for the same figure, axis label “ng D. virilis” is too casual; something like “Concentration of D. virilis RNA (ng)” would be more appropriate. In addition, it would also be visually much more informative if the graph proportions were adjusted: since the objective of the normalization was to make the slope 1, a square graph would illustrate this point much more easily. Figures 1B-E have no y-axis labels, and the x-axes as well as axes numerals are not readable at all. Overall the resolution of the figures is also somewhat low, but I think this will probably be adjusted before publication. Similar complaints about text size and missing axes labels apply for Figure 2. Figure 3 also needs descriptive axes labels, not just “Slope 1”, “Slope 2”. In this figure, the plot area should be expanded so that the legend is not obscuring some of the data points at the top right. Finally, the figure referencing in the text is incorrect. (What is Figure 2.2?). All in all, the figures and their captions should be modified such that they are largely understandable on their own, without having to read extensively into the text.

Experimental design

The experimental design is appropriate, however, it would be helpful for the authors to provide more primary data such as that shown in Figure 1A, but for the other methods they tested as well, perhaps as part of supplemental data.

Validity of the findings

In Figure 1, the authors comment: “The mean correlation coefficient was statistically and practically indistinguishable between the Clontech samples and the SMART-seq2 samples (t-test p = .11, Figure 2.2).” and in Figure 1 show the “Distributions of slopes, intercepts, and correlation coefficient for linear regressions of the abundance of each gene”. I noticed that in these figures, although the mean of each is very similar, as the authors pointed out, the distributions of each method do display differences. For example, Clontech and TotalScript methods show a tail skew for the slopes and intercepts distributions, whereas the distribution of these same metrics for TruSeq shows symmetry. Can the authors comment on this difference, or provide additional data and or speculation as to why this might be or the origin of this difference?

Authors should provide additional detail in the bioinformatics methods – were raw reads quality filtered/controlled in any way prior to mapping?

Comments for the author

No additional comments.

·

Basic reporting

The article deviates from the PeerJ suggested format of "Intro, Methods, Discussion, Conclusion" and instead uses "Intro, Results, Methods" with "Experiments" enumerated within Results. I find this somewhat disorganized, particularly in the semi-arbitrary distinction between "Experiments" described in results and "Methods", particularly since the article is fundamentally a comparison of several methods. I think the standard Intro/Methods/Discussion format would communicate the information more clearly.

The introduction and abstract should more clearly define the scope as pertaining to linear response of RNA-seq measurement. Many RNA-seq experiments rely on other important properties of the data such as representation of transcript direction and uniformity of coverage (i.e. Levin, 2010) which are not explored here and are not even possible with some of the low input methods described. Library diversity (another relevant metric) is not explored in the dataset at hand but is used as justification (line 72) for the study.

Figure 1A is very suspicious; the region around 10 ng D. virilis is difficult to interpret as a distribution, and it is odd that 10ng and 20ng are so broad while the region around 16ng appears unusually tight. I suspect this is a graphing artifact and so may be misleading.

The axes labels in all distributions are too small to be legible.

There are other minor edits noted in the attached PDF.

Experimental design

Overall, the experimental design and analysis reasonably documents the method comparison with respect to linearity of gene expression measurement. The introduction should more clearly state this scope particularly since many other RNA-seq comparison papers use many other metrics.

The simulation data is an excellent component of this paper.

The method used to establish the "practical lower input limit" of the TruSeq kit is based on two data points in a single replicate being at or below the lower limit of detection of a common but not sensitive assay (the Nanodrop). Neither the method ("failures") nor the measurement technique (Nanodrop) are sufficient to establish a lower limit. This should either be re-assessed using a Qubit assay, qPCR, or BioAnalyzer, or restricted in conclusion to "anecdotal".

Other clarifications are noted in the attached PDF.

Validity of the findings

With a few exceptions, the findings are robust, statistically sound, and controlled.

The justification of 70 ng as a conservative lower limit is not well justified. Since it is only 30% lower than the manufacturers' stated lower limit and, as stated in the text, this is still several orders of magnitude higher than practical for single-cell experiments, the authors might re-consider simply using the manufacturer's lower limit and present the anecdotal data as simply that.

An exploration of read depth is described in the introduction (lines 63/64) but is only touched on briefly in the simulation section. This analysis does not constitute "the effect of read depth on the quality of the data."

While the error may have been minor, the "pooling error" needs more explanation to understand whether this was an error in quantitation or simply in volume measurement. This is important to preserve the integrity of the rest of the experimental data.

There are several instances the authors make personal appeals (e.g. the phrase, "we believe") (lines 8, 91, 153, 218, 223). These should either be supported by data or citation, deleted, or rephrased as assertion/assumption.

---

## Round 0.2 · accepted · Accept

· Academic Editor

Accept

Thank you for addressing the reviewer comments.

·

Basic reporting

No additional comments.

Experimental design

No additional comments.

Validity of the findings

No additional comments.

Comments for the author

No additional comments.